# Resolving the conformational ensemble of a membrane protein by integrating small-angle scattering with AlphaFold

**Samuel Eriksson Lidbrink**[ID][1], **Rebecca J. Howard**[1,2], **Nandan Haloi**[ID][2]*, **Erik Lindahl**[ID][1,2]*

**1** Science for Life Laboratory, Department of Biochemistry and Biophysics, Stockholm University, Solna, Sweden, **2** Science for Life Laboratory, Department of Applied Physics, KTH Royal Institute of Technology, Solna, Sweden

* haloi@kth.se (NH); erik.lindahl@dbb.su.se (EL)

**Data availability statement:** Scripts for performing the workflow and applying it to

## Abstract

The function of a protein is enabled by its conformational landscape. For non-rigid proteins, a complete characterization of this landscape requires understanding the protein's structure in all functional states, the stability of these states under target conditions, and the transition pathways between them. Several strategies have recently been developed to drive the machine learning algorithm AlphaFold2 (AF) to sample multiple conformations, but it is more challenging to a priori predict what states are stabilized in particular conditions and how the transition occurs. Here, we combine AF sampling with small-angle scattering curves to obtain a weighted conformational ensemble of functional states under target environmental conditions. We apply this to the pentameric ion channel GLIC using small-angle neutron scattering (SANS) curves, and identify apparent closed and open states. By comparing experimental SANS data under resting and activating conditions, we can quantify the subpopulation of closed channels that open upon activation, matching both experiments and extensive simulation sampling using Markov state models. The predicted closed and open states closely resemble crystal structures determined under resting and activating conditions respectively, and project to predicted basins in free energy landscapes calculated from the Markov state models. Further, without using any structural information, the AF sampling also correctly captures intermediate conformations and projects onto the transition pathway resolved in the extensive sampling. This combination of machine learning algorithms and low-dimensional experimental data appears to provide an efficient way to predict not only stable conformations but also accurately sample the transition pathways several orders of magnitude faster than simulation-based sampling.

GLIC, as well as the AF-generated conformations, are available at https://doi.org/10.5281/zenodo.13692051. The experimental SANS data were retrieved from the Small-Angle Scattering Biological Data Bank (SASBDB), https://www.sasbdb.org/project/1317/ (resting conditions from [SASDL33] and activating conditions from [SASDL43]).

**Funding:** This work was funded by grants from the Swedish Research Council (VR) to E.L. under contracts 2019-02433 & 2021-05806, and from the Swedish e-Science Research Center to E.L. and R.J.H. S.E.L was supported by Sven and Lilly Lawski's PhD fellowship. N.H. was supported by a Marie Sklodowska-Curie Postdoctoral Fellowship (grant no. 101107036). AlphaFold2 runs were performed using the Berzelius resource funded by the Knut and Alice Wallenberg Foundation (project no. Berzelius-2023-244). The funding agencies had no role in study design, data collection and analysis, decision to publish or preparation of the manuscript.

**Competing interests:** The authors have declared that no competing interests exist.

## Author summary

The dynamic behavior of proteins is key to their function, including nerve signaling, enzyme catalysis, and cellular regulation. These functions rely on precise movements and shape changes that allow proteins to interact with other molecules. Understanding protein structures and their evolution at the atomic level is thus crucial for many applications such as drug development, but remains a challenging problem. High-resolution experimental techniques can determine the structural states of many proteins, but often struggle to capture less-populated states. While computational approaches can model protein dynamics, they can be expensive and are typically limited to short time scales that may not encompass the full range of biologically relevant behavior. Recently, artificial intelligence-driven tools like AlphaFold2 (AF) have emerged to predict protein structures with high accuracy. However, they usually default to predicting a single structure, and while modified workflows allow for sampling of alternative states, it can be difficult to assess their functional relevance. Here, we introduce a method that combines AlphaFold2 with small-angle scattering data to predict multiple protein states and their frequencies under specific biological conditions. This approach offers a computationally efficient alternative for integrating experimental data with computational methods, providing a new tool for studying protein dynamics.

## Introduction

Numerous proteins, ranging from channels, transporters, and receptors to different enzymes, achieve their biological functions by undergoing specific sequences of conformational transitions, where each state has unique properties e.g. for binding or conductance [1–4]. To characterize these functions, we need an understanding of the structural details of these functional states, as well as the transition pathways involved. Although high-resolution experimental methods such as X-ray crystallography and Cryo-electron microscopy have had immense success in determining the most populated structures of many proteins [5–7], they are often inefficient in capturing multiple functional states and are rarely able to determine the relative population of the different states.

Low dimensional in-solution experimental methods on the other hand, such as small-angle scattering (SAS), nuclear magnetic resonance (NMR) spectroscopy, and Förster resonance energy transfer (FRET), can in principle capture information from all conformational states with non-negligible population [8,9], although the resolution is occasionally low for short-lived states. Resolving structural information about the actual conformational transition is even more challenging since the intermediate states do not have significant populations under equilibrium conditions. Computational approaches provide attractive options to study this type of dynamics, and limitations related to model accuracy and accessible timescales can often be addressed by combining them with low-resolution experimental data [10,11]. However, while these methods have successfully solved many different problems — e.g. by combining molecular dynamics (MD) simulations with SAXS [12,13], single-molecule FRET [14], NMR and neutron reflectometry [15] and NMR and cross-linking mass spectrometry [16] — the associated simulations often require substantial computational costs.

A potentially more computationally efficient approach for generating structural models of proteins is to use machine learning-based methods such as AlphaFold2 (AF2) [17], RoseTTA fold [18], or AlphaFold3 [19]. While largely successful, these methods usually default to predicting the single most stable conformation. Interestingly, some recent studies have

combined AF2 with SAS data, utilizing different simulation techniques to capture structural variation from the single predicted state [20–22]. As an alternative approach, it has been shown that stochastically subsampling the multiple sequence alignment (MSA) depth can drive AF2 to sample alternative stable conformations [23–25]. While subsampling will generate alternative predictions, it can also lead to increasing fractions of incorrect structures, and it is sometimes unclear what conformations are physiological and what conditions they correspond to unless there are already experimental structures available that can be used for direct comparison. Similarly, since information about the full energy landscape is typically not available even for test cases, it is difficult to assess to what extent the methods correctly resolve diversity, relative populations, and transition intermediates.

One protein where such an assessment is valuable is the proton-gated cation channel GLIC, a pentameric channel from the prokaryote *Gloeobacter violaceus*, which is also a good test case with extensive amounts of data available. GLIC poses interesting challenges due to its size (with $5\times \sim 320$ residues), subtle gating transitions (with $C\alpha$ RMSD $\approx 2.7$ Å between its closed [PDB ID 4NPQ] and open [PDB ID 4HFI] states), and the fact that the majority of GLIC conformations are expected to remain closed even under activating conditions [26,27]. A particular advantage of GLIC is the availability of several experimentally determined structures in both open and closed states, and we have recently performed small-angle neutron scattering (SANS) experiments to capture conformational information at room temperature [27], and extensive molecular dynamics (MD) simulations combined with Markov State Modeling that resolve the entire free energy landscapes of the transitions [26]. The former provides experimental data that is specific to particular conditions, while the latter provides a highly valuable ground truth (within the limits of force field accuracy) to assess the accuracy of the order-of-magnitudes faster machine-learning predictions.

Here, we propose a method for integrating low-dimensional experimental data - in particular SAS - with MSA-based AF-sampling of alternative states (Fig 1) to identify multiple functionally relevant structural states of a given protein and determine their relative population, and apply it to the GLIC protein. The method stochastically subsamples the MSA depth to allow AF to sample multiple conformations of a given protein, and then calculates the theoretical SAS intensity profile of all AF-sampled conformations to select high-confidence scoring conformations with distinct SAS intensity curves. Finally, corresponding experimental data is used to determine the population and viability of the different selected states under multiple conditions. Based on the crystal structures and the free energy landscapes of GLIC, it is possible to thoroughly assess the quality of the predictions, both in terms of the selected states, their relative populations, and sampled intermediate states.

## Results

### Theoretical SANS curves of AF-sampled conformations robustly define discrete clusters

To sample structurally diverse conformations of the GLIC protein, we generated 960 GLIC conformations using AF by stochastically subsampling the MSA depth (see Methods for details). While most AF-generated conformations seemed physically plausible by visual inspection, a handful of conformations appeared misfolded. We observed that conformations with average predicted local distance difference test (pLDDT) scores [17,28] < 75, in general, had poor structural quality when assessed by metrics such as MolProbity score [29] and the number of steric clashes and cis-amide bonds [30] (S1 Fig). Therefore, as an initial quality filter, we removed all conformations with average pLDDT scores below 75, leaving 874 conformations. Many of these conformations were structurally similar, with root mean square

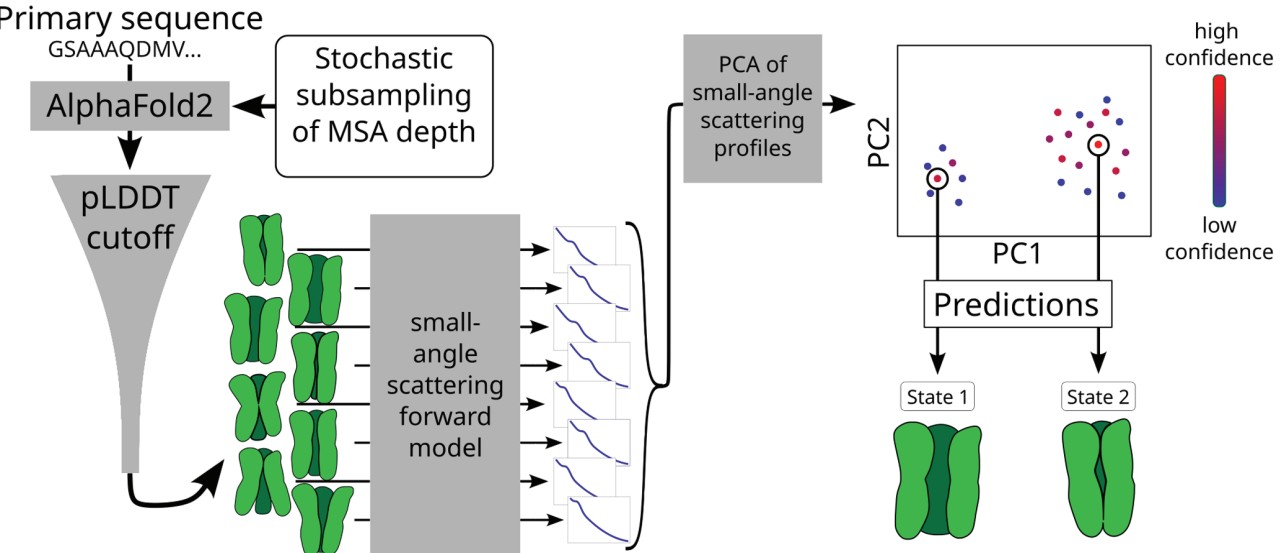

**Fig 1. A schematic depiction of the method for combining AlphaFold2 with small-angle scattering data to predict key conformational states of proteins.** An ensemble of conformational states (in green) is generated from AlphaFold2 by stochastically subsampling the multi-sequence alignment (MSA) depth. The theoretical small-angle scattering (SAS) intensity profile is calculated for each conformation, followed by Principal Component Analysis (PCA) to project these profiles into a lower-dimensional space. In the PC space, conformations are clustered and the highest confidence-scoring conformations from each cluster are chosen as predictions of the protein states.

deviation (RMSD) of the C$\alpha$ atoms within 1 Å of the average structure in the ensemble. 133 conformations (15 %) deviated more than 1 Å from the average structure, and 17 deviated more than 2.5 Å (S2 Fig), suggesting that multiple functional states might be represented. Still, some members of even this filtered ensemble featured implausible geometry (S3 Fig). We determined that an alternative analysis was needed to appropriately filter nonphysical structures and distinguish plausible functional states.

To this end, we assessed our ensemble of AF-generated models according to low-resolution structural information accessible by SAS. Since SAS data cannot be inverted to directly predict structures, we instead predicted theoretical curves for all GLIC conformations in our ensemble, using the implicit solvent method implemented by Pepsi-SANS [31]. By performing principal component analysis (PCA) of these profiles, and projecting onto the first two principal components (PCs) (which accounted for more than 95% of the total variance), this initially filtered ensemble appeared to represent a single cluster (Fig 2A). However, by progressively restricting the number of conformations in the PCA by raising the pLDDT cutoff, we found that the data increasingly split into two clusters (S1 Video). We hypothesized that distinct clusters could correspond to distinct physiological states of GLIC, and sought to find an appropriate pLDDT score cutoff. For this purpose, we performed agglomerative clustering for different cutoff values and calculated the average silhouette scores of the clusterings (see Methods for details). The average silhouette score increased from 0.46 (indicating poor cluster separation) for the initial pLDDT cutoff of 75, to a maximum of 0.79-0.81 (reflecting more distinct cluster separation) for pLDDT cutoffs of 86.5–87.2 (S4 Fig). At the latter cutoffs, two distinct clusters were visually clear (Fig 2B), and we selected the conformations with the highest average pLDDT score in each cluster as the best-quality models. Interestingly, we only observed two distinct clusters when filtering this system based on average pLDDT scores, not on pTM or ipTM scores (S4 Fig, S1 Video).

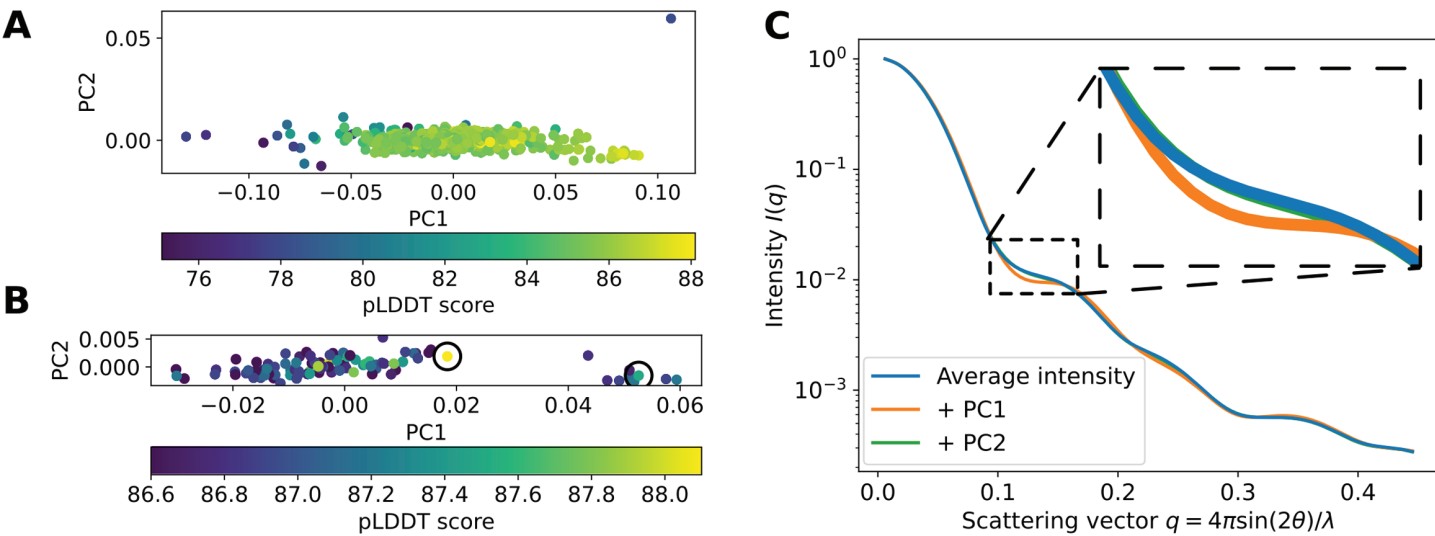

**Fig 2. Theoretical SANS curves of AF-sampled conformations separate into two distinct clusters.** (A, B) Principal component analysis (PCA) of the theoretical SANS profiles of all AF-generated conformations with an average pLDDT score above 75 (A) and 86.6 (B). Black circles in B indicate the conformations in each cluster with the highest pLDDT scores: 88.1 (left) and 87.5 (right). (C) Average SANS curve of the AF-generated conformations with pLDDT≥ 86.6 (in blue), as well as the curve when the first or second principal component from panel B is added to the average (in orange and green, respectively). The SANS curves of the PCs are scaled by the maximal value of the corresponding PC coordinates in B.

To visualize the experimental relevance of the PCs described above, we calculated the average SANS curve for our pLDDT-filtered conformational ensemble, then compared it to theoretical curves in which the maximum value of a given PC is added to this average (Fig 2C). The variations in PC1 corresponded to clear differences in intensities for some scattering vectors $q$, especially for $q \in [0.10, 0.17] =: \mathcal{Q}$ (while the variation in PC2 was relatively small). Seemingly small changes at intermediate $q$-values are critical in SANS to differentiate between different conformational states of a protein, since the intensities at the lowest $q$-values relate mostly to the radius of gyration of the protein (which may remain rather constant across different states), while the intensities at the highest $q$-values may contain substantial noise. Encouragingly, we also found that the range $\mathcal{Q}$ corresponded to the range $q \in [0.10, 0.12]$ where the experimental SANS data sets differed the most for the different pH conditions [27], indicating that the variation captured by PC1 was experimentally relevant.

To further assess what types of states were sampled, we calculated the pore radii of the two predicted states using CHAP [32]. The pore was relatively constricted for the state corresponding to a lower value along PC1 (Fig 2B, S5 Fig), with a minimum radius of 1.7 Å. In contrast, the state associated with a higher value along PC1 (Figure 2B, SI Figure S3) had a minimum radius above 2.2 Å, approaching that of a hydrated sodium ion. Therefore, we approximated the former state as a closed state, and the latter as a representative open state.

Since this approach depends on the ensemble of conformations generated by AF, which in turn depends on the random seed used, we evaluated the robustness of the method by re-running the entire pipeline five additional times with different random seeds, resulting in five new sets of AF-generated conformations. By again filtering based on pLDDT scores, we were able to assign clusters with maximal average silhouette scores of 0.78, 0.75, 0.79, 0.64 and 0.64, respectively (S6 Fig). In four cases, conformations selected from each of two

clusters could be assigned as representative closed or open states according to the protocol above (S5 Fig). In a single case, both conformations contained equivalently constricted pores, suggesting the approach is generally applicable to distinguishing functional states, and that at least some limitations can be resolved with increased sampling.

## Change in experimental SANS profiles corresponds to the difference between generated states

Next, we sought to evaluate the selected open- and closed-state models using experimental SANS data. While the implicit solvent-based SANS forward model makes it possible to perform the pipeline quickly, the calculation includes several hyperparameters usually fitted to the experimental data. Earlier studies have indicated that the fitting of these hyperparameters might affect the prediction quality and could reduce the information that can be extracted from the experimental data [33,34]. To more accurately capture subtle variations in the SANS curves, the explicit-solvent-based method WAXSiS implemented in GROMACS-SWAXS [35,36] was used to calculate SANS curves of the selected conformations (see Methods). These results were compared to experimental SANS curves for GLIC collected using a paused-flow size-exclusion chromatography setup under pH 7.5 (resting) and pH 3.0 (activating) conditions, collected at 10°C for 58 and 53 min with average concentrations of 0.77 and 0.74 mg/ml, respectively [27](Fig 3A, 3B).

Given that we were mainly interested in the relative changes between the two states, rather than absolute properties of either state, we focused our analysis on characterizing whether the

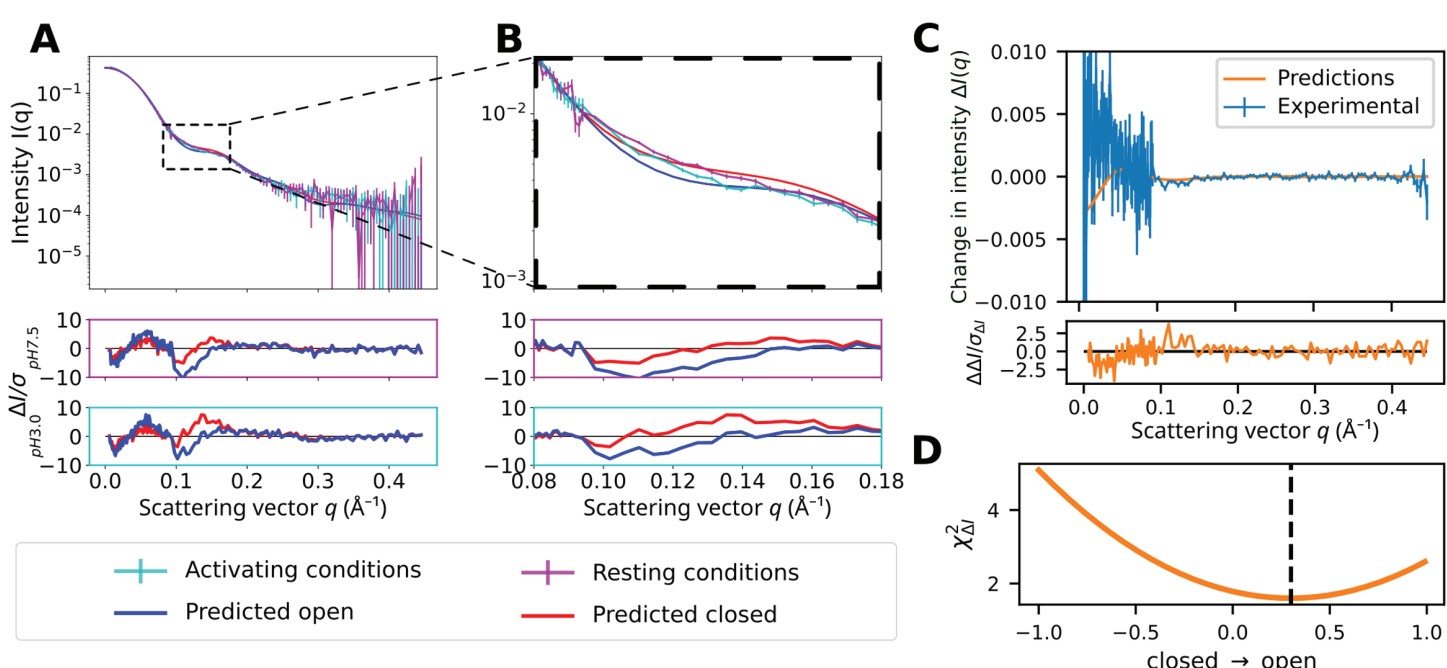

**Fig 3. Change in experimental SANS profiles corresponds to the difference between generated states.** (A, B) Theoretical and experimental SANS curves at resting (pH 7.5) and activating (pH 3.0) conditions of the predicted GLIC conformational states for scattering vectors $q \in [0, 0.45]$ Å$^{-1}$ (A) and $q \in [0.08, 0.18]$ Å$^{-1}$ (B). Error-normalized residuals are shown below. (C) Experimental and theoretical SANS difference curves between activating and resting conditions as well as between the open and closed prediction, with the latter scaled to fit the former. An error-normalized residual is shown below. (D) Fit $\chi^2_{\Delta I}$ of the predicted versus experimental difference curves as a function of the population shift from the closed prediction to open prediction. The dashed line indicates the optimal fit, and corresponds to a 30% increase in the contribution of the predicted open versus closed states.

difference between the predicted states could account for the observed difference between experimental conditions (Fig 3C). This allows characterization of potentially relevant substructure transitions, even in the context of possible systematic differences in computed vs. experimental models. Encouragingly, the difference between the explicit-solvent SANS-curves was the largest at the scattering vectors where the experimental SANS curves differed the most (Fig 3A, 3B). An optimal fit to the difference curve between activating and resting conditions ($\chi^2_{\Delta I} \approx 1.6$) was achieved by a 30% increase in the contribution of the predicted open versus closed states (Fig 3D). That is, if the population under resting conditions were entirely closed, the population under activating conditions should be 30% open. This estimate is consistent with previous reports that a minority population of GLIC transitions from the closed to open state when pH is decreased from 7 to 3 [26,27,37].

Fits derived from our additional replicate AF ensembles were comparable to those of the initial test case. Specifically, for the four replicates in which distinct functional states were predicted, the population shift that best fit the difference curve between experimental conditions involved an increase in predicted open versus closed states of 49%, 29%, 23%, and 27% respectively, with $\chi^2_{\Delta I} \in \{1.6, 1.6, 1.7, 1.7\}$ (S7 Fig).

We next examined the direct fits to the experimental SANS curves, which provide an additional way of estimating the state populations. For all relevant replicates, the best fit to resting conditions (with $\chi^2 \in \{3.6, 2.2, 2.7, 4.4, 4.5\}$) was achieved by the predicted closed conformation, while the best fit to activating conditions (with $\chi^2 \in \{4.6, 6.0, 3.3, 4.1, 5.1\}$) was achieved by linear combinations of 92/8 %, 36/64 %, 100/0 %, 100/0 %, 100/0 % closed/open conformations (S8 Fig, Fig 3C, 3D). To characterize the quality of these predictions, we sought to compare them to approximate ground truth models derived from extensive molecular dynamics (MD) simulations [26] (described in more detail below). We found that the MD-sampled conformations representing the closed-state basin individually fit the experimental SANS data slightly better, with $\chi^2$-values of $2.0 \pm 0.3$ (mean $\pm$ standard deviation) under resting conditions and $2.0 \pm 0.5$ under activating conditions, as determined by the implicit solvent method Pepsi-SANS [31]. To investigate the discrepancies in the fits from the AF-sampled conformations, we calculated the corresponding residuals, which exhibited a clear q-dependent bias that was similar for both models (Fig 3A, 3B). For a more intuitive view, we compared the experimental SANS data and the predicted intensities in real space (using the tool "calculating pair distance distribution functions for proteins", CaPP) [38]). Both the predicted models appeared to be modestly contracted relative to experimental SANS profiles, although the differences are subtle (S9 Fig). Interestingly, the population estimates based on the experimental difference curves appeared to be less affected by these discrepancies than the direct fits.

## Predicted states are consistent with crystal structures

The approach described above enables reproducible predictions of discrete conformations of GLIC without using existing structural data. These predictions were consistent with functional data, with a closed state primarily corresponding to resting conditions and an open state associated with activating conditions. To examine whether the predictions reasonably represent the open and closed states, models were compared to crystal structures reported to represent closed (PDB ID 4NPQ) and open (PDB ID 4HFI) states of GLIC (Fig 4A, 4B, S5 Fig]) [39]. The predicted conformations were consistent with corresponding crystal structures (with C$\alpha$ RMSD values of 1.59 Å and 2.09 Å for the open and closed states respectively when aligning on the C$\alpha$-atoms), especially in the transmembrane domain (TMD)

(with C$\alpha$ RMSD values of 0.68 Å and 0.78 Å respectively). The pore radius profiles are highly similar for the closed states and they exhibit qualitatively similar expansions in the upper region (2.4 Å for predictions vs. 3.6 Å for crystal structures) (S5 Fig). The fits to the crystal structures were similar for the conformations obtained when re-running the pipeline (detailed RMSD values can be found in S1 Table).

Gating in the pentameric ligand-gated ion channel family features characteristic structural transitions, which can be assessed in the context of the ensemble of AF-generated conformations, and the specific predictions. Channel opening in GLIC has been associated with an expansion of the distance between the upper regions of the pore lining M2 helix and the center of mass of the pore (referred to as "M2 spread", following the nomenclature from [26]) (Fig 4C), as well as a contraction of the ECD ("ECD upper spread") (Fig 4) [26,40]. In relation to these metrics, most of the AF-generated conformations appeared closed-like, while sampling of open-like conformations was sparse, despite that the majority of the GLIC structures deposited in the PDB are open [41]. Still, the predictions both of the closed and open states featured similar M2 spread and ECD upper spread as the corresponding crystal structure, suggesting the pipeline identified reasonable models of both states (Fig 4D, 4F, S10 Fig).

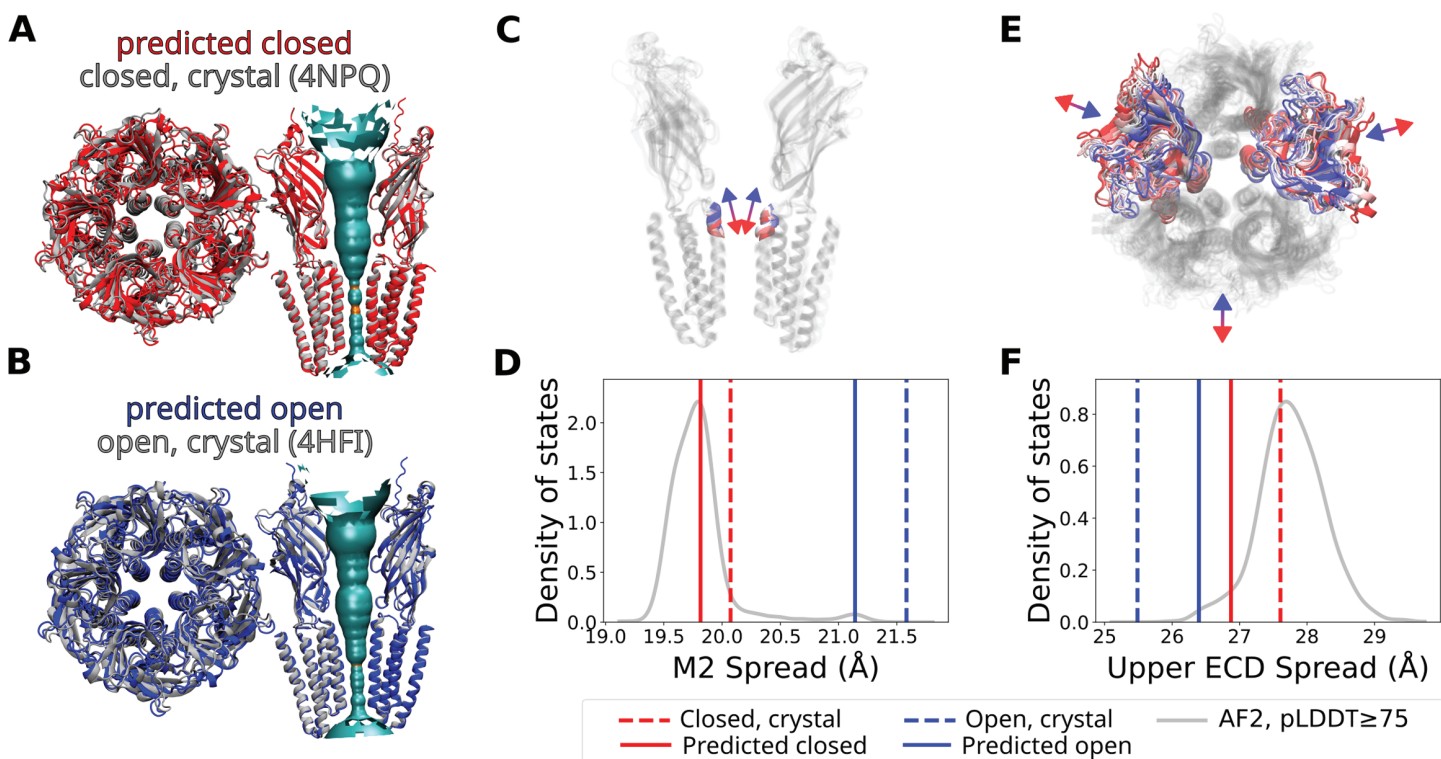

**Fig 4. AlphaFold2-generated conformations overlap well with crystal structures of the open and closed state of GLIC.** (A, B) Overlay of experimentally determined crystal structures in (A) the closed state (PDB ID 4NPQ) and (B) the open state (PDB ID 4HFI) with the corresponding prediction (visualized using VMD [42]). The structures were aligned to minimize the RMSD of all C$\alpha$ atoms in both the predicted and corresponding crystal structures. The pore hydration profiles for the predicted structures (calculated using HOLE [43]) are also shown, with cyan and orange signifying wide (radius > 2.3 Å) and narrow (radius < 2.3 Å) parts of the pore, respectively. (C) Distance between the centers of mass of the pore and that of the upper part of the pore lining M2 helix (M2 spread) for AF-generated conformations ranging from closed-like (in red) to open-like (in blue). (D) The corresponding values for the predictions and the crystal structures as well as the density of states for all AF-generated conformations with an average pLDDT score above 75. (E, F) Upper spread of the extracellular domain (ECD), depicted as in panels C and D, respectively.

## AF-sampled ensembles map to low-energy regions of MD-simulated free-energy landscape

Ideally, generative models should produce ensembles sampling not only equilibrium conditions, but also conformations along transition pathways. Such non-equilibrium conformations are difficult to assess directly from experimental data since they do not have a significant population at equilibrium conditions, but an attractive feature of the GLIC model system is the availability of an approximate ground truth in the form of physics-based simulations. For this purpose, we used the free-energy landscapes calculated from extensive (120 μs) molecular dynamics (MD) simulations of GLIC combined with Markov state modeling (MSM) under resting and activating conditions, as approximated by fixed protonation of pH-titratable amino-acid residues [26]. These free-energy landscapes have been constructed using time-lagged independent component analysis (tICA) to capture the system's free energy as described by its slowest collective degrees of freedom.

There is a remarkably good overlap between the AF-sampled ensemble of structures and the free-energy landscapes (Fig 5, S11 Fig). While some conformations with low average pLDDT scores projected onto regions corresponding to non-physical high-energy states (under both resting and activating conditions), the filtering based on pLDDT scores successively removed virtually all such conformations (S2 Video). Most conformations with pLDDT ≥ 75 projected onto regions with free energies less than 2 kcal/mol in at least the activating conditions (Fig 5). Interestingly, these projections ranged continuously from closed-like to open-like states.

It is not certain that transitional conformations in tIC space represent functional transition states, as they could deviate along other degrees of freedom orthogonal to the first two components. Still, particularly in light of its narrow RMSD range, our AF-generated ensemble appeared to follow the predicted transition path, with spreads corresponding to the wells and

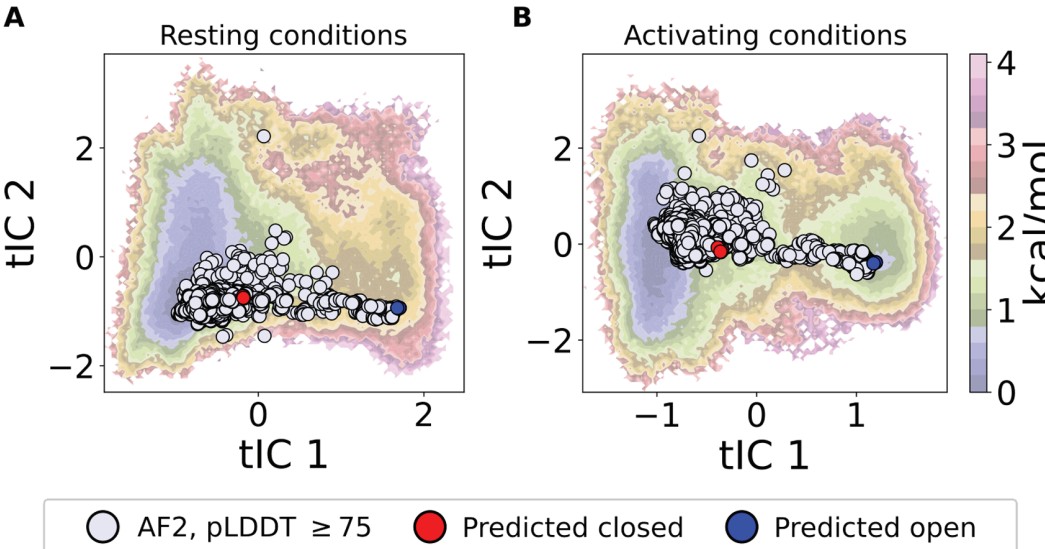

**Fig 5. The predicted states are consistent with free energy landscapes from MD simulations.** Projections of the AF-generated conformations with pLDDT ≥ 75 onto the deprotonated (A) and protonated (B) energy landscape of GLIC, with predicted closed and open conformations marked in red and blue respectively. For the protonated energy landscape (B) the free energy well on the left (where tIC1∼ −1) corresponds to closed conformations of GLIC while the free energy well on the right (where tIC1∼ 1.5) corresponds to open GLIC conformations [26].

saddle points in the free-energy landscape (Fig 5B). MD-generated ensembles sampled generally greater root-mean-square fluctuations (RMSFs) than our AF ensemble, particularly in loops between strands or helices; still, the profile of low versus high flexibility was consistent between methodologies (S12 Fig).

The selected predictions of the open and closed channel states agree similarly well with the corresponding local free energy minimum energy basins at both resting and activating conditions, particularly for the open state (Fig 5, S13 Fig), while the ensemble also appears to reproduce the broader diversity of the closed state observed in simulations and experiments.

## Discussion

A proper characterization of conformational ensembles is key to understanding the function of many proteins. Here, we have introduced a method for combining MSA-based AF-sampling of alternative states with SANS data, and applied it to the pentameric ion channel GLIC. This approach identifies distinct conformational states of GLIC and combined with experimental SANS data it is possible to accurately estimate the relative population of the distinct states under different specific experimental conditions. Further, the AF-sampled states project onto regions of the MD-calculated free-energy landscapes that continuously followed the transition pathway between the open and closed state of the channel. To put the computational efficiency of the machine-learning approach into perspective, it required approximately 36 GPU hours to sample the 960 conformational states used in the initial run in this study, while it required around 1.7 million CPU hours to run the corresponding MD simulations for each experimental condition. Notably, the AF-SANS approach *does not* in any way rely on the MD simulations for predictions - it is only used as a ground truth to validate the accuracy of the ensemble prediction.

For a general use case, this approach makes it possible to select one or multiple target biological conditions and, after collecting SANS or SAXS data (or other low-dimensional data that depends on conditions), identify different conformational states and determine the relative population of the identified states under the targeted condition(s). While this study is restricted to the GLIC ion channel, we expect the approach to be applicable to diverse sets of proteins with multiple functional states.

The subtle gating transitions and a low open probability of GLIC only resulted in minor differences in the experimental SANS curves [27], but we could still robustly estimate the population shift between the experimental conditions that largely agreed with expectations (Fig 3). Given the subtlety of the differences, the usage of explicit-solvent SANS forward models may have been particularly important, especially since the vestibule is largely hydrated in both the closed and open states.

Some analytical choices in this work, for example in the clustering of SAS curves and selection of representative conformations, may be system-dependent. In the case of GLIC, SANS curves predicted for our AF-generated models consistently formed two distinct clusters once filtered by an appropriate pLDDT cutoff, where intermediary states had been filtered out. However, average pLDDT scores are thought to indicate general model quality, not necessarily thermodynamic stability [44]. Moreover, a recent study has shown that pLDDT scores can be biased toward structures already published in the PDB, and may not correlate strongly with model quality for alternative conformations [45]. The usefulness of pLDDT as a screening metric may thus be system-specific. One approach to tackle this is to try multiple AF metrics like pLDDT, pTM and ipTM, as well as structure quality metrics like MolProbity, and select empirically a filtering protocol that identifies distinct clusters. Alternatively, it may be more useful to filter conformations to an acceptable level of geometric plausibility, and then choose

representative conformations that are distinct in the SAS PCA space, even if intermediary conformations are still present.

The usefulness of combining machine learning with experimental data depends on the ability of forcing AF to sample alternative conformational states as well as the accuracy of the sampled prediction. In the case of GLIC, it was possible to sample both open- and closed-like states. We cannot exclude the possibility that AF is influenced by deposited PDBs of open and closed GLIC proteins. Interestingly, however, AlphaFold sampled mostly closed-like conformations (Fig 4, S11 Fig), even though a substantial majority of GLIC structures in the PDB are open [41]. This suggests that the distribution of AF-generated conformations is not heavily biased by the distribution of structures in the PDB. Instead, the distribution might depend on the structural details of the different states like the number of contacts formed, suggesting that this method is generally applicable even when one state is unknown. Still, MSA-subsampling might fail to sample alternative states for other proteins (e.g. as seen in [46]). Other methods have recently been developed to improve the AF-sampling of alternative states, e.g. by manipulating the MSAs through in-silico mutagenesis (SPEACH_AF) [47], using dropout at inference (AFsample) [48], clustering the MSAs (AF-Cluster) [44] or stochastically masking the columns in the MSAs (AFsample2) [49], and could trivially be integrated into this method as alternatives. It is also possible that the predictions could be improved by using AlphaFold3 [19] instead of AlphaFold2, although it remains to be seen to what extent MSA-subsampling and the other MSA-based techniques enable AF3 to sample alternative states.

Still, a recent study highlighted that both AF2 and AF3 have difficulties in sampling all states of fold-switching proteins, suggesting that other algorithms might be needed to yield sufficient sampling of diverse states in such cases [50]. In general, the method described here is not limited to AlphaFold, and could be used to combine any machine learning algorithm, that has some minimal form of sampling of the functional states with low-dimensional experimental data.

Our predicted open and closed models were consistent with previously reported X-ray structures, and approached corresponding basins in free-energy landscapes from our previous extensive MD simulations [26]. Still, MD-sampled conformations representing the closed-state basin fit the experimental SANS data slightly better than our AF-based models. It is perhaps unsurprising that models derived from extensive statistical sampling of MD simulations are somewhat better optimized to solution-phase structural data than AF2 models, which may be influenced by contact optimization and/or structural training data.

With machine learning algorithms rapidly improving, the potential loss in accuracy is constantly decreasing and in particular, given the dramatic reduction in computational cost, it should be possible to compensate by simply increasing sampling or using short simulations to improve occupancy predictions after the initial ensemble prediction.

Here, we represented the conformational ensemble of GLIC using only two states. This is an oversimplification, and given the imperfect fits to the SANS data, it might be appropriate to include more AF-sampled states. Indeed, our AF ensemble sampled a similar pattern of RMSF as our previous MD-generated ensembles (S12 Fig); on the other hand, it did not replicate either the density of states or quantitative flexibility (Fig 5, S11 Fig). Since AF sampling is not statistically representative, it is not straightforward to include more conformational states without running the risk of overfitting to noise in the SANS data. To attain a Boltzmann-distributed ensemble of protein states, one approach could instead be to use the AF-sampled conformations as seeds for MD simulations e.g. in line with [24]. Since the AF-sampled states seemingly ranged continuously from open- to closed-like for GLIC (Fig 5), it is plausible that simulations starting from these seeds could reconstruct the free-energy landscape.

Predictions from improved machine learning algorithms still benefit from the inclusion of low-dimensional data, e.g. by providing population estimates at target conditions. While this study showcases the usefulness of SAS, it could also be generalized to other types of low-dimensional data. A recent study has for instance shown that one could combine SPEACH_AF with double electron-electron resonance data to illuminate the energy landscape for alternating access of bacterial homologs of neurotransmitter sodium symporters [51]. The combination of machine learning algorithms with low-dimensional experimental data can thus provide high-resolution structures and additional information such as population estimates at low computational costs, complementing conventional computational methods as well as high-resolution experimental techniques.

## Methods

### Ensemble generation

AlphaFold2 conformations were generated by running AlphaFold-Multimer [52] in Colab-Fold [53], where all the MSAs are obtained from the MMseqs2 database [54]. The MSA depth was subsampled according to methods proposed by del Alamo et al. [23], using a single recycle and no energy minimization. The sequence number parameters were selected by initially generating conformations with $max\text{-}extra\text{-}seq \in \{32, 64, 128, 256, 512, 1024\}$ and $max\text{-}seq = max\text{-}extra\text{-}seq/2$ using all five different AlphaFold-Multimer models ($num\text{-}models$=5) with one seed per depth. As the conformations at depths 32 and 64 appeared unfolded, we focused our search at $max\text{-}extra\text{-}seq \geq 100$. Conformations were generated with $max\text{-}extra\text{-}seq = 100, 120, \dots, 380, 400$ for all five AlphaFold-Multimer models using 12 random seeds per depth and model, resulting in a total of 960 conformations. The parameter $random\text{-}seed$ was set to $60n$ for rerun $n \in \{1, 2, 3, 4, 5\}$.

### Generation of SANS intensity profiles using the implicit solvent model

PEPSI-SANS [31] was used to calculate the scattering intensity curves for all generated conformations. No experimental data was used, the number of points set to 5000, intensity $I(0) = 1$, hydration shell contrast $\delta\rho = 5.50898\%$ (default), no protein deuteration and buffer deuteration 100% (to mimic experimental conditions in [27]), and the effective atomic radius of solvent displacement $r_0$ to $r_0/r_m = 1$ (default) (where $r_m$ is the average atomic radius over the protein).

### Clustering of theoretical SANS intensity profiles

Clustering was performed on each set of AF-generated conformations with average pLDDT scores above a given threshold, ranging from 75 to the maximal pLDDT score in the ensemble, including all intermediate values. For each set, we used PCA to project the corresponding theoretical SANS intensity curves onto the $n \geq 2$ first principal components accounting for more than 95 % of the total variance from the mean. Agglomerative clustering was then used to label all conformations in the set, with the number of clusters set to 2. Based on this clustering, we calculated the silhouette scores for all filtered conformations. The pLDDT threshold for which the clustering had the highest average silhouette score was chosen as the optimal threshold for clustering. At this threshold, we selected the highest average pLDDT scoring conformations from the two corresponding clusters as predictions of conformational states of GLIC.

## Explicit solvent SANS calculations

The explicit solvent SANS calculations were computed from restrained MD simulations of the selected AF-generated conformations. The MD simulations were performed using GROMACS [55] v2024.1 utilizing the AMBER99SB-ILDN force field [56]. Each conformation was solvated using the SPC/E water model [57] in a cubic box neutralized with 150 mM NaCl, energy minimized, followed by 10ns production runs. The protein was positionally restrained during the simulations using force constants of 1000 kJ mol$^{-1}$ nm$^{-2}$ to preserve the protein's conformational state for the SANS calculations. Long-range electrostatic was treated using the particle mesh Ewald (PME) method [58] with default settings. Van der Waals interactions used a force-based smoothing function to 1.2 nm, with neighbor list settings using the default GROMACS auto-tuning. Water molecules were constrained by using the SETTLE algorithm [59], while hydrogen bond lengths were restrained with LINCS [60], using a default 2 fs time step. The Nose-Hoover thermostat [61] was used to keep the system at 303.15 K.

SANS curves were computed from the MD simulations using the WAXSiS method [35,36] implemented by GROMACS-SWAXS [62], by following the workflow outlined in [62]. In these calculations, the distance of the spatial envelope to the protein surface was set to 7 Å, the solvent density of the water was set to 334 e nm$^{-3}$ and 1500 scattering vectors **q** per absolute value of **q** were used for orientational averaging.

## Comparing theoretical SANS intensities with experimental data

The goodness of fit of the theoretical SANS intensities $I_1(q_i)$ and $I_2(q_i)$ (computed from the two selected conformations) for a given set of corresponding weights $w_1$ and $w_2$ (where $w_1 + w_2 = 1$) compared to the experimental data $I_{\text{exp}}(q_i)$ (at either activating or resting conditions) with experimental errors $\sigma(q_i)$ over $N_q$ scattering vectors $q_i$, was calculated by minimizing

$$\chi^2 = \sum_{i=1}^{N_q} \frac{\left[ f \cdot (w_1 I_1(q_i) + w_2 I_2(q_i)) - I_{\text{exp}}(q_i) \right]^2}{N_q \cdot \sigma^2(q_i)}, \tag{1}$$

with respect to the scaling parameter $f$. We did not include a fitting term sometimes used to compensate for insufficient background subtraction, to avoid overfitting to noise.

To compute the change in experimental intensity for some scattering vectors $q$ that differed slightly ($<10^{-4}$ Å$^{-1}$) between the experimental conditions, we linearly interpolated the experimental data under activating conditions to match the scattering vectors under resting conditions, with error bars determined through error propagation. For the predicted change in SANS curves we used the average $\bar{f}$ of the scaling parameters $f$ that minimized Eq (1) for the activating and resting conditions, and computed the fit $\chi^2_{\Delta I}$ of the predicted vs. experimental difference curves as

$$\chi^2_{\Delta I} = \sum_{i=1}^{N_q} \frac{\left[ \bar{f} \cdot (\delta w I_1(q_i) - \delta w I_2(q_i)) - (I_{\text{exp}}^{\text{active}}(q_i) - I_{\text{exp}}^{\text{resting}}(q_i)) \right]^2}{N_q \cdot (\sigma^2_{\text{active}}(q_i) + \sigma^2_{\text{resting}}(q_i))}, \tag{2}$$

where $\delta w$ signifies the change in weights from prediction 2 to 1, $I_{\text{exp}}^{\text{active}}$ and $I_{\text{exp}}^{\text{resting}}$ the experimental SANS intensities under activating and resting conditions respectively, and $\sigma_{\text{active}}$ and $\sigma_{\text{resting}}$ the corresponding experimental errors.

### SANS curves of MD-sampled closed-state representatives

To select representative conformations of the closed-state basin, we randomly sampled 1550 conformations from the resting (deprotonated) MD ensemble [26] and chose the 277 conformations that had free energies below 0.5 kcal/mol. The corresponding SANS curves were calculated using (the implicit solvent method) Pepsi-SANS [31], with $I(0)$, $\delta\rho$ and $r_0$ optimized for the best fit to the respective experimental curves.

## Supporting information

**S1 Fig. Low plDDT scores correlate with poor structural quality.** (A) MolProbity scores [29] of all the AF-generated conformations as a function of the corresponding plDDT score. (B, C, D, E) count of (B) non-planar amide bonds, (C) cis proline amide bonds, cis amide bonds and steric clashes, all calculated using the TopModel program [30], for each AF-generated conformation as a function of the corresponding plDDT score.
(TIF)

**S2 Fig. Variability in the AF-sampled ensemble.** Probability density of the root mean square deviation of the C$\alpha$ atoms of the AF-generated conformations with pLDDT $\geq 75$, compared to the average structure. Alignment is done on the C$\alpha$ atoms.
(TIF)

**S3 Fig. Example of a poorly folded AF-generated conformation with an average plDDT score of $\sim 78$.** The protein conformation was visualized using ChimeraX [63].
(TIF)

**S1 Video. Filtered projections of SANS curves using different AF-quality metrics.** SANS PCA space for conformations with pLDDT, pTM or ipTM scores above an increasing cutoff. Shown for the initial run of the pipeline.
(MP4)

**S4 Fig. Cluster separation for different AF-quality metrics.** Average silhouette score of the agglomerative clustering of the SANS intensity profiles of all conformations with pLDDT (top left), ipTM (top right), or pTM (bottom) scores above different cutoffs for the initial run of the pipeline, as a function of the number of such conformations. The dashed red line indicates the cutoff for the maximal silhouette score.
(TIF)

**S5 Fig. Pore profiles of predicted conformations and crystal structures of the open (4HFI) and closed (4NPQ) states.** Grey arrows indicate the maximal pore-radii expansions between the predicted conformations (solid) and the crystal structures (dashed). The pore profiles were calculated using CHAP [32].
(TIF)

**S6 Fig. Clustering quality across the reruns of the pipeline.** Average silhouette score of the agglomerative clustering of the SANS intensity profiles of all conformations with pLDDT scores above different cutoffs for the five reruns, as a function of the number of such conformations. The dashed red line indicates the cutoff for the maximal silhouette score.
(TIF)

**S7 Fig. Fit of the change in SANS intensity.** Fit $\chi^2_{\Delta I}$ of the predicted intensity difference to the experimental as a function of the population shift from the closed prediction to open

prediction, shown for the four reruns in which distinct functional states were predicted. The dashed line indicates the optimal fit.
(TIF)

**S8 Fig. Fits of selected conformations to experimental SANS data.** Fits to the experimental SANS data for a linear combination of the closed and open prediction as a function of their relative weights, for the initial run as well as the four reruns in which distinct functional states were predicted.
(TIF)

**S9 Fig. Pairwise distance distribution of the selected conformations compared to the experimental data.** The distributions are normalized by the area under the respective curve.
(TIF)

**S1 Table. C$\alpha$ RMSD of the predicted structures compared to experimental crystal structures.**
(PDF)

**S10 Fig. M2 spread and upper ECD spread for the reruns.** Distance between the centers of mass of the pore and that of the upper part of the pore lining M2 helix (M2 spread) and the upper spread of the extracellular domain (upper ECD spread) for the predicted structures, the crystal structures, as well as the density of states for all AF2-generated conformations with an average pLDDT score above 75. All data is shown for the four reruns in which distinct functional states were predicted.
(TIF)

**S11 Fig. Probability density distribution of the AF2 conformations projected onto the free energy landscapes.** Shown for the initial pipeline run projected onto resting conditions (left) and activating conditions (right).
(TIF)

**S2 Video. Free energy landscape projections for increasing pLDDT cutoffs.** AF-generated conformations with pLDDT scores above an increasing threshold projected to the free energy landscape at activating conditions. Shown for the initial run of the pipeline (top left) and the reruns.
(MP4)

**S12 Fig. Root mean square fluctuation (RMSF) of the AF-generated conformations and the MD-generated ensembles.** Subunit average of the RMSF for the MD-ensembles as well as the AF-generated conformations (A) projected onto a snapshot from the ensembles and (B) plotted as a function of residue number. The MD-ensembles were retrieved from [26], and the molecular models were visualized using ChimeraX [63].
(TIF)

**S13 Fig. Free energy landscape projections for selected conformations.** Projections of the AF2-generated conformations with pLDDT $\geq$ 75 onto the free energy landscape of GLIC at resting and activating conditions, for the four reruns in which distinct functional states were predicted.
(TIF)

## Author contributions

**Conceptualization:** Nandan Haloi.

**Data curation:** Samuel Eriksson Lidbrink.

**Formal analysis:** Samuel Eriksson Lidbrink.

**Funding acquisition:** Erik Lindahl.

**Investigation:** Samuel Eriksson Lidbrink.

**Methodology:** Samuel Eriksson Lidbrink.

**Project administration:** Rebecca J. Howard, Erik Lindahl.

**Resources:** Nandan Haloi.

**Software:** Samuel Eriksson Lidbrink.

**Supervision:** Rebecca J. Howard, Nandan Haloi, Erik Lindahl.

**Validation:** Samuel Eriksson Lidbrink.

**Visualization:** Samuel Eriksson Lidbrink.

**Writing – original draft:** Samuel Eriksson Lidbrink.

**Writing – review & editing:** Samuel Eriksson Lidbrink, Rebecca J. Howard, Nandan Haloi, Erik Lindahl.

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
