## [Decision Letter · Decision Letter 0]

13 Dec 2024

PCOMPBIOL-D-24-01584

Resolving the conformational ensemble of a membrane protein by integrating small-angle scattering with AlphaFold

PLOS Computational Biology

Dear Dr. Lindahl,

Thank you for submitting your manuscript to PLOS Computational Biology. After careful consideration, we feel that it has merit but does not fully meet PLOS Computational Biology's publication criteria as it currently stands. Therefore, we invite you to submit a revised version of the manuscript that addresses the points raised during the review process.

Please submit your revised manuscript within 60 days Feb 12 2025 11:59PM. If you will need more time than this to complete your revisions, please reply to this message or contact the journal office at ploscompbiol@plos.org. Please include the following items when submitting your revised manuscript:

We look forward to receiving your revised manuscript.

Kind regards,

Joanna Slusky, Ph.D.

Academic Editor

PLOS Computational Biology

Jason Papin

Editor-in-Chief

PLOS Computational Biology

**Additional Editor Comments (if provided):**

Please focus your revisions on the key critiques of the reviewers.

**Journal Requirements:**

We ask that a manuscript source file is provided at Revision. Please upload your manuscript file as a .doc, .docx, .rtf or .tex.We noticed that you refer to data points not shown in the legend of S1 Fig. We do not allow references to unshown data, as the PLOS data access policy requires that all data be either published with the manuscript or made available in a publicly accessible database. Please amend the supplementary material to include the referenced data or remove the references.For main figures, please upload separate figure files in .tif or .eps format.For more information about figure files please see our guidelines: https://journals.plos.org/ploscompbiol/s/figureshttps://journals.plos.org/ploscompbiol/s/figures#loc-file-requirements

**Reviewers' comments:**

Reviewer's Responses to Questions

**Comments to the Authors:**

Reviewer #1: Numerous studies have shown that it is sometimes possible to use AF2 to predict conformational ensembles, but selecting functionally relevant states remains a challenge. In this work, Lindbrink and colleagues combine AF2 modeling with small angle neutron scattering data to estimate the relative populations of open/closed GLIC conformations under resting and activating conditions. Their modeling approach aims for rigor and generalizability. Some of the details raise concerns, however.

Major:

1. The fits of theoretical curves (generated from AF2 models) to experimental curves do not align very well, especially under activating conditions (chi-squared ~4.5). It is also concerning that fits from replicate AF ensembles varied widely under activating conditions: from 0-64% open. Both the relatively high chi-squared values and the wide variance in results suggests that this form of modeling might not be robust enough to describe the relative populations of open and closed conformations under activating conditions, seemingly one of the main objectives of this manuscript. Could more robust results be obtained if the differences between curves were compared? IE are the relative trends between differences in experimental data and differences in calculated curves were similar, could that provide a more robust result?

2. The efficiency of using AF2 over MD simulations is appreciated. Still, how do the fits of low-energy MD conformations compare with SANS experiments, and what does that say about the usefulness of AF2-based modeling? This question seems especially important because the predicted closed AF2 structures do not appear to overlap with the energy minima in Figure 5.

3. The clustering approach by plDDT seems a bit overdone. Looking at Figure 2A, by eye, there seem to be 3 clusters (a large one centered around 0.0, a second centered on 0.4, a third on 0.6). There does not seem to be a strong justification to increase plDDT threshold, especially given the next comment.

4. Several recent papers have shown that plDDT scores do not necessarily discriminate between correct and unphysical AF2 predictions (PMIDs: 37981824, 39181864, Bryant and Noé, Nat. Comms. 2024). Since the work in this article separates different conformations by plDDT (Fig 2), can the authors comment further on when the method they propose is likely to be successful and when it may not be?

Minor:

1. Another manuscript (Riccabona, et al bioRxiv 2024) suggests that AF2’s ability to replicate MD ensembles is limited. How can that be reconciled with the results presented here?

Reviewer #2: The manuscript by Eriksson Lidbrink et al. describes a possibly interesting approach of AlphFold-based conformational sampling guided by small-angle scattering data. However, the manuscript is premature and poorly written, hence I am not able to review it in the current form. As soon as the language and the statements have matured, I am willing to review the manuscript. I stopped reading the manuscript at line 40. In the current form, I can only recommend rejecting the manuscript.

Please find below a list of imprecise statements and sloppy language up to line 40, which precluded me from reviewing this potentially interesting study.

Significance statement:

- line 2: What is a "precise movement"? A movement into a precisely defined states?

- line 4: What is meant with "evolution" of a protein structure? The same as "movement"?

Introduction:

Line 12: "Low dimensional experimental methods" - what is meant with this? Data may be low-dimensional, but methods?

Line 25: "although the resolution is occasionally low for short-lived states". Is the resolution higher for long-lived states? The word "occasionally" is imprecise and is not helpful for the reader? Methods like SAS and NMR do not even report a "resolution".

Line 19: "this type of dynamics" is unspecific.

What is meant with "model accuracy"? Force field inaccuracies?

Line 20: "combining them" - "them" is unspecific.

Line 21: "while these methods have successfully solved many different problems" - what is meant with "these methods", the "computational approaches", or their combination with low-resolution experimental data? Which "problems"?

Line 26: "A potentially more computationally efficient approach". Why "potentially"? AF is more efficient than MD simulations.

Line 28: "While largely successful" - successful for what, by which measure?

Line 29: "Interestingly, some recent studies have" - "Interestingly, some" does not transport any useful information here.

Line 34: "While subsampling will generate alternative predictions, it can also lead to increasing fractions of incorrect structures" needs a reference or other type of support.

Line 35: "what conformations are physiological and what conditions" is sloppy, should be "which conformations" and "which conditions"

Line 36: "full energy landscape" - complete energy landscape?

Line 39: "what extent the methods correctly resolve diversity, relative populations" - "what extent" is sloppy. What is meant with "diversity", diversity of what, by which measure?

Reviewer #3: see attachment

Reviewer #4: The authors address a very important question in the field, namely how to combine AF-subsampling with experimental data and how well the subsampled conformational states align with extensive MD data.

I think the story is well written and the analysis is timely. I have three remarks that should be addressed before publication.

First, the authors mention that they excluded (quality filtered) structures based on low pLDDT scores. Another way to check if conformations are plausible are to check for D-amino acids/cis-amid bonds or sterically clashes.

Did the authors consider to check for those inaccuracies? I would be curious to see whether low pLDDT scores correlate with physically incorrect/implausible structures? (https://www.tandfonline.com/doi/pdf/10.1080/19420862.2023.2175319
https://www.nature.com/articles/s42003-023-04927-7), also in line with that, did AF subsampling pick up areas of highest flexibility compared to the MD? Could the authors use in addition to the tICA like a B-factor plot overall comparing the subsampled ensemble with the MD ensemble to compare areas of flexibility?

Also did the authors observe if the relative predicted populations resulting from subsampling correspond to the number of structures available in the PDB of the respective states?

While I know this question might be out of scope for this study I would be curious to hear the authors comment/ add a paragraph in the discussion, if using the AF subsampled conformations as starting structures for MD simulations would be enough to sample and reconstruct the free energy surface obtained from aggregated 120 µs of sampling (in line with workflows shown in these following papers: https://pubs.acs.org/doi/10.1021/acs.jctc.2c01189, https://www.sciencedirect.com/science/article/abs/pii/S0969212624003708).

**Have the authors made all data and (if applicable) computational code underlying the findings in their manuscript fully available?**

Reviewer #1: **No: **I was not able to download the supplement from the website. I got it from bioRxiv.

Reviewer #2: None

Reviewer #3: Yes

Reviewer #4: None

PLOS authors have the option to publish the peer review history of their article (what does this mean?). If published, this will include your full peer review and any attached files.

Reviewer #1: No

Reviewer #2: No

Reviewer #3: No

Reviewer #4: No

**Figure resubmission:**
---

## [Decision Letter · Decision Letter 1]

11 Apr 2025

PCOMPBIOL-D-24-01584R1

Resolving the conformational ensemble of a membrane protein by integrating small-angle scattering with AlphaFold

PLOS Computational Biology

Dear Dr. Lindahl,

Thank you for submitting your manuscript to PLOS Computational Biology. After careful consideration, we feel that it has merit but does not fully meet PLOS Computational Biology's publication criteria as it currently stands. Therefore, we invite you to submit a revised version of the manuscript that addresses the points raised during the review process.

Please submit your revised manuscript within 30 days Jun 11 2025 11:59PM. If you will need more time than this to complete your revisions, please reply to this message or contact the journal office at ploscompbiol@plos.org. Please include the following items when submitting your revised manuscript:

We look forward to receiving your revised manuscript.

Kind regards,

Joanna Slusky, Ph.D.

Academic Editor

PLOS Computational Biology

Jason Papin

Editor-in-Chief

PLOS Computational Biology

**Additional Editor Comments (if provided):**

**Journal Requirements:**

**Reviewers' comments:**

Reviewer's Responses to Questions

**Comments to the Authors:**

Reviewer #1: The authors have addressed my concerns well. Congratulations on the nice manuscript.

Reviewer #3: The authors have made changes to their manuscript in response to the reviewers comments and included ensemble, residuals, and real space comparisons, as requested. The current figures better visualize computational and experimental. On looking at the experimental vs prediction figures, gains made are minimal. Based on crystallographic RMSD, the authors have accurately predicted protein structures that are already deposited in the PDB. Based on SANS, the agreement experimental and models are open to interpretation and could be argued. The overall agreement of the curves to the models are good – as one would expect when the PDB includes the open and close structures, but the question is about differentiating two states. The authors write "The approach described above [comparison to SANS] enables accurate prediction of discrete conformation of GLIC." where discrete conformations are open and close states based on SANS data. Interpreting accuracy as matching the experimental data, the authors have not shown this clearly. Being able to accurately match predictions with solution data is an important problem, and the discussion section addresses this issue. However, the method here appears to be the first step and not the final one. Details are shown below.

1. The accuracy of predicted models to experimental data is not clear.

a. Figure 3 shows visible bias in the residuals. The similarity of 3A/B red and blue curves suggest that whatever is wrong is wrong in both sets of models.

b. Figure 3C shows the difference between open and close for experimental and open and close for predicted, as requested by one of the reviewers. It is not clear to me the information value of this plot. As an analogy, 5000-4999=2-1.That doesn't mean 5000 and 2 are similar to each other.

c. Figure 3D shows the chi square fit trying different ensembles, which is the closest I could find to a comparison between experimental and predicted. However, following a single metric that has bias for lower q does not justify the statement for "accurate prediction".

d. Based on Fig. 3B, the major difference between open and close experimental is in the mid q range between 0.1 and 0,2, and this is where the emphasis should be placed.

e. Supplemental Fig 10 provides the real space calculation of the experimental and predicted. The experimental differences between open and close are subtle. By eye, the difference between the best predicted and either experimental curve is greater than the difference between experimental curves. The curves are difficult to distinguish, so if one peak is narrower than the other, it is difficult to judge.

f. It is difficult to conclude, based on the SANS analysis, that the authors were able to accurately predict discrete conformations. SANS could serve as a validation for the GLIC model, but not with regard to the open and close states. The authors may want to consider that this discrepancy between predicted and experimental may in part be due to the large pore even in the closed state and possible PEPSI-SAXS error in the the hydration layer prediction within that pore The authors may consider gromacs, which is reported to better handle hydration.

2. For the crystallographic comparison, it is also difficult to see clearly. The overlays act like clouds in Fig 4 A, B, and E, and whether those clouds show differences between open and close is up for interpretation. The models have different pore sizes but does that reflect the pore size of the crystal is not obvious. The RMSD S11 table is the closest, but it is a singular value for an atomic structure. The most important aspect of the open and close is the pore size. It would be good to include a plot of the HOLE pore radius as a function of height (as viewed by A and B) for best predictions and crystal.

3. With regard to potential influence from the PDB, the authors state that the prevalence of open states in the PDB vs closed states in the AF prediction shows that AF is not biased. This reasoning doesn't make sense to me, 1) if AF is trained on the crystallographic database as a whole, which represents primarily lower energy states. The occurrence of closed states in the activated conditions, as determined in this paper, suggests that the closed state is the lower energy state. 2) How do you know that one example is sufficient for it to remember the relative relationship of residues in the closed state. To address this issue, the authors should consider adding in something like "we cannot exclude the possibility that AF is influenced by deposited PDBs of open and closed GLIC proteins"

Reviewer #4: The authors have addressed all my comments. I recommend for publication!

**Have the authors made all data and (if applicable) computational code underlying the findings in their manuscript fully available?**

Reviewer #1: None

Reviewer #3: Yes

Reviewer #4: Yes

PLOS authors have the option to publish the peer review history of their article (what does this mean?). If published, this will include your full peer review and any attached files.

Reviewer #1: No

Reviewer #3: No

Reviewer #4: **Yes: **Monika Fernández-Quintero

**Figure resubmission:**
---

## [Editor Report · Decision Letter 2]

30 May 2025

Dear Lindahl,

We are pleased to inform you that your manuscript 'Resolving the conformational ensemble of a membrane protein by integrating small-angle scattering with AlphaFold' has been provisionally accepted for publication in PLOS Computational Biology.

Best regards,

Joanna Slusky, Ph.D.

Academic Editor

PLOS Computational Biology

Jason Papin

Editor-in-Chief

PLOS Computational Biology

---

## [Editor Report · Acceptance letter]

PCOMPBIOL-D-24-01584R2

Resolving the conformational ensemble of a membrane protein by integrating small-angle scattering with AlphaFold

Dear Dr Lindahl,

I am pleased to inform you that your manuscript has been formally accepted for publication in PLOS Computational Biology. Your manuscript is now with our production department and you will be notified of the publication date in due course.

With kind regards,

Zsofia Freund
